

**Geospatial Analysis and Simulation of Glacial Lake Outburst Flood**
**Hazard in Hunza and Shyok Basins of Upper Indus Basin**
Syed Naseem Abbas Gilany[1]*, Javed Iqbal[2] and Ejaz Hussain[3]
[1]School of Civil and Environment Engineering; naseemphd13@igis.nust.edu.pk
[2]School of Civil and Environment Engineering; javed@igis.nust.edu.pk
[3]School of Civil and Environment Engineering; ejaz@igis.nust.edu.pk
*Correspondence: naseemgillani2000@yahoo.com; Tel.: +92-321-519-6790
**Abstract:** The UIB (Upper Indus Basin) is prone to GLOFs (Glacial Lake Outburst
Floods) because of host of factors encompassing global warming in general and
anthropogenic in particular. Physical monitoring of such a large area on regular basis is a
challenging task especially when the temporal and spatial extent of the hazard is highly
variable. The purpose of this study was to identify the potentially dangerous glacial lakes and
simulate the associated hazard to the downward settlements using HEC-RAS in the GIS
environment by utilizing Landsat 7 remote sensing data. The study was conducted in Hunza
and Shyok sub-basins of UIB where there are several human settlements which are
endangered due to GLOFs hazard. Sudden breaches in the unstable moraine dams adjoining
receding glaciers had hardly been simulated previously for rapid and huge accumulation of
turbulent water in the glacial lakes. The ASTER DEM (Digital Elevation Model) is utilized
in this study to detect flow accumulation of glacial hazard involving slope, elevation, and
orientation of the mountain glaciers. A historic glacial outburst had affected settlements of
Hunza valley, and destroyed the village of Passu almost 40 kms from the outlet of the glacial
lake. The infrastructure including houses, buildings and farmlands in Hunza and Shyok
basins remains threatened because of GLOFs hazard. This study proposes a need to establish
a concrete scientific context which can be ultimately fed to more encompassing predictive
frameworks for early warning of potential hazards in the area. The study contributes in
simulation of potential hazardous GLOFs of Hunza and Shyok basins in order to earmark
hazard extents and to generate an early warning to the habitation. The study results revealed
that settlements of Hunza and Shyok basins are threatened by the GLOFs hazard. Keeping in
view the seasonal growth of the potentially dangerous glacial lake of Hunza basin, a low
discharge of 3500 $m^3$/s from potentially dangerous glacial lake can affect 40%, whereas, a
moderate discharge of 5000 $m^3$/s can affect 60% and a high discharge of 7000 $m^3$/s can affect
80% of the Shimshal village habitat. In Shyok basin, a low discharge of 100 $m^3$/s from
potentially dangerous glacial lake can affect 20%, whereas, a moderate discharge of 300 $m^3$/s
can affect 30% and a high discharge of 500 $m^3$/s can affect 40% of the Barah village habitat.
The results of this study provide a platform for the establishment of an early warning and
monitoring system to minimize the impact of future GLOFs which has been a grey area.
Accurate and comprehensive simulation of potentially dangerous GLOFs is of utmost
importance for risk assessment. A digital repository of GLOFs hazard extents can enhance
the ability to inform policy makers on the vulnerability, risk mitigation and action/adaptation
measures.
**Keywords:** Anthropogenic, Disastrous; Geo-Morphology, Glacial Lake Outburst Flood.



## 1. INTRODUCTION

The alpine glaciers of the sub-continent region are a renewable natural freshwater storehouse that benefits hundreds of millions of people downstream (Shah and Kanth, 2013). Owing to global warming acceleration, the glaciers of the mid-latitude region of Pakistan are retreating since the second half of the 20th century (Das and Meher, 2019). On the retreating glacier terminus, this phenomenon has accounted for the accumulation of many disastrous glacial lakes. The damming by unstable moraines has caused several glacial lakes. The disastrous GLOFs containing debris and a large quantity of turbulent water lead to the sudden breaches of these unstable moraines which hold huge quantities of water (Bhambri et al., 2013).Glaciation and inter-glaciation are natural processes that have occurred several times during the last 10,000 years. A situation that provides a large space for retaining melt water, leading to the formation of moraine-dammed lakes (Che et al., 2014). Glacier-connected lakes have likely accelerated the glacial retreat via thermal energy transmission and contributed to over 15% of the area loss in their connected glaciers. On the other hand, significant glacial retreats led to disconnections from their proglacial lakes, which appeared to stabilize the lakes in the Himalayas. Continuous expansions in the lakes connected with debris-covered glaciers, therefore, need additional attention due to their potential outbursts (Nie et al., 2018). Glacier retreat is an indication of glacial lake formation. The glacial hazard of GLOFs can cause loss of life, livestock, property, valuable forests, costly mountain infrastructures, farmlands, and pasture resources. Damages to settlements and farmland can take place at very great distances from the outburst source, for example, in Pakistan, a damage occurred 1,300 Km from the outburst source (Gilany and Iqbal, 2019). Much of the damage created during GLOF events is associated with the large amounts of debris that accompany the floodwaters (Budhathoki et al., 2010). In the past 20 years, glaciers in the Himalayas have retreated and thinned rapidly as a response to regional climate warming, leading to the formation of new glacial lakes and the expansion of existing glacial lakes (Kaushik et al., 2019). These areas are located in the border belt and the Eurasian plates, where tectonic seismic activity is frequent and intense. Earthquakes have often compromised the stability of mountain slopes, glaciers, and moraine dams, resulting in an imbalance in the state of glacial lakes (Wang and Zhou, 2017). During recent decades there has been a rapid retreat of glaciers all over the world, new lakes are being formed, and the size of the existing lakes attached to the glaciers is increasing. Another emerging hypothesis of more GLOF events is the change in the pattern of rainfall (Khan et al., 2019, Harrison et al., 2018). A historic glacial flood burst had a depth of around 30 m at the junction of Shimshal and Hunza, (about 40 Km from the assumed position of the lake) and destroyed the village of Passu near the Hunza river (Goudie, 1984). Glacier-fed lakes are dominant in both quantity and area and exhibit an overall faster expansion trend compared to the non-glacier-fed lakes in the Himalayas (Mir et al., 2018). Formation of glacial lake phenomenon has rapidly increased owing to global warming. Glaciological characteristics of the ablation zone of Baltoro glacier have changed in the recent past because of a host of factors (Mayer et al., 2006). The diversity of glacial material is the prime reason for the diverse behavior and peculiar dynamics of the glaciers. Heterogeneity in the Karakoram glacier surges is observed because



of the peculiar dynamics of the glaciers (Quincey et al., 2015). The glacier surges are
propagated coupled with glacial lakes. Glacier changes in the Karakoram region are mapped
temporally in order to observe the diverse behavior of the glaciers (Rankl et al., 2014). On
debris-covered glaciers, glacial lake formation is observed at a faster pace in alpine region of
Pakistan (Hambrey et al., 2008, Raup et al., 2007). A conceptual analysis model of
supra-glacial lake formation on debris-covered glaciers is based on GPR (Ground
Penetrating Radar) (Mertes et al., 2017). The analysis has put forth the argument of increased
melting observed in glaciers of northern Pakistan. The risk factor increases exponentially
with the presence of supra-glacial lakes, the trend is observed through modeling and risk
assessment of GLOFs (Lala, 2018). The frequency of glacier-dammed lakes and outburst
floods in the Karambar valley of Hindukush-Karakoram has increased the risk to
infrastructure and living organisms in this region (Iturrizaga, 2005). The glacier surge is a
seasonal phenomenon owing to the extreme flow velocities resulting in the formation of a
dammed lake (Steiner et al., 2018). The balance in accumulation and ablation zones of a
glacier is very vital for its stability. A hydro-meteorological perspective on the anomaly of
glacier dynamics has originated the argument of heavy accumulation zones, thus disturbing
the mass balance (Bashir et al., 2017). The natural stability and behavior of the glacier are
very much dependent on slope, elevation, aspect, and geomorphology of the vicinity. Glacier
expansion is very much related to the elevation from mean sea level in the Karakoram region
(Hewitt, 1998). Himalayan glaciers are a focus of public and scientific debate. Prevailing
uncertainties are of major concern because some projections of their future have serious
implications for water resources. Most Himalayan glaciers are losing mass at rates similar to
glaciers elsewhere; except for emerging indications of stability or mass gain in the
Karakoram (Bolch et al., 2012). Rising global temperature is the major factor in the glacial
lake formation which is caused by the glacial retreat in mountainous regions. In the era from
1550 to 1850, the glaciers were quite in length in comparison with today. With the inception
of global warming, moraines formation adjacent to glaciers blocks the glacial lakes
(Bhutiyani, 1999). Since the Little Ice Age, it is said that the glaciers of the Himalaya have
experienced a retreat of approximately one kilometer in length. A situation leads to the
formation of moraine-dammed lakes with the provisioning of a large space for melt water
retention (Mool et al., 2001). The proximity analysis of the settlements with respect to glacial
lakes is very vital with respect to geospatial analysis and modeling of GLOFs hazard in
Hunza and Shyok basins of upper Indus basin. Nearly 35 devastating GLOF events have
occurred during the last 200 years in Gilgit Baltistan (Din et al., 2014). The frequency and the
intensity of GLOF events has risen over the past few years according to available records.
During the three decades the glacier cover been decreased an average of 10.1% which cause
many GLOF in the Hunza and Shyok (Ali et al., 2019). Also during the year (2008-2009) five
GLOF events took place in Hunza Valley (Kreutzmann et al., 2011). Study of the GLOF
Events shows that such an event has been associated with weather condition in terms of
temperature increase, precipitation and heat waves.
Keeping in view the fact that watersheds of Pakistan are covered by major glaciers,
which are quite susceptible to disastrous outbreak / flooding hazards, the objectives of the
study were (i) to map potentially dangerous glacial lakes in Hunza and Shyok basins and (ii)



to simulate the flood extents of the potentially dangerous glacial lakes using HEC-RAS
model and do damage assessment to the downstream settlements in these basins.

## 2. MATERIALS AND METHODS

### 2.1 Geomorphology

High mountains of Pakistan comprise the western end of 2,400 km long Himalayan
range and some parts in the Hindukush and Karakoram ranges. Northern areas spread over
72,496 km2 with a midst towering snow-clad peaks having heights varying from nearly
1,000 to over 8,000 meters above sea level. Of the 14 over 8,000 m peaks on earth, 4 occupy
an amphitheater at the head of Baltoro glacier in the Karakoram Range. These are: K-2
(Mount Godwin Austen) which is 8,611 m and is world second highest peak, Gasherbrum-I
(8,068 m), Broad Peak (8,047 m) and Gasherbrum II (8,035 m). In addition to these, there are
68 peaks over 7,000 m and hundreds which are over 6,000 m high. Generally, because of
their rugged topography and the rigors of the climate, the northern highlands and the
Himalayas to the east have been formidable barriers to movement into Pakistan throughout
history (Isserman et al., 2010).

### 2.2 Climate

Pakistan is basically a dry country of the warm temperate zone. The climate of the area
is transitional between that of central Asia and the monsoonal region of south Asia, which
varies considerably with latitude, altitude, aspect and local relief. There is not only high
spatial variability but temporal variability is quite high as well. Except for a small strip of
sub-tropical terrain in Punjab and the wet zone on the southern slopes of the Himalayan and
Karakoram mountain ranges, most of the country is arid or semi-arid steppe land. The
snowmelt run-off constitutes a substantial part of water resources of the rivers of Pakistan
(Singh et al., 2011). The Indus River, primarily supplied by glaciers in its upper reaches, and
subject to the least seasonal variation, still has a maximum flow more than fifty times its
minimum. Alpine glaciers contribute 50% of the Indus water flow. The Indus River is about
2,800 km long and 62% of its catchment lies in Pakistan (Singh et al., 2011). The swelling of
Indus and its tributaries is subjected to volumetric decrease of glaciers and if coupled with
heavy monsoonal rains, can cause floods during summer (Gilany and Iqbal, 2018).

### 2.3 Glaciated River Basins of Pakistan

For hydrological studies, Pakistan's northern area is divided into 10 major river basins
(Figure 1). Clockwise from west, these basins are of Swat River, Chitral River, Gilgit River,
Hunza River, Shigar River, Shyok River, Indus River, Shingo River, Astor River, and the
Jhelum River. Most of the snow and ice reserves are concentrated in the mountain ranges
lying in these basins (Gillani, 2014, Ashraf et al., 2015). These basins contain glaciated part,





which forms headwaters of the main Indus basin. The study area encompasses the Hunza and
Shyok basins of alpine glaciers of Pakistan (Ali and De Boer, 2007, Kääb et al., 2012,
Bookhagen and Burbank, 2010).

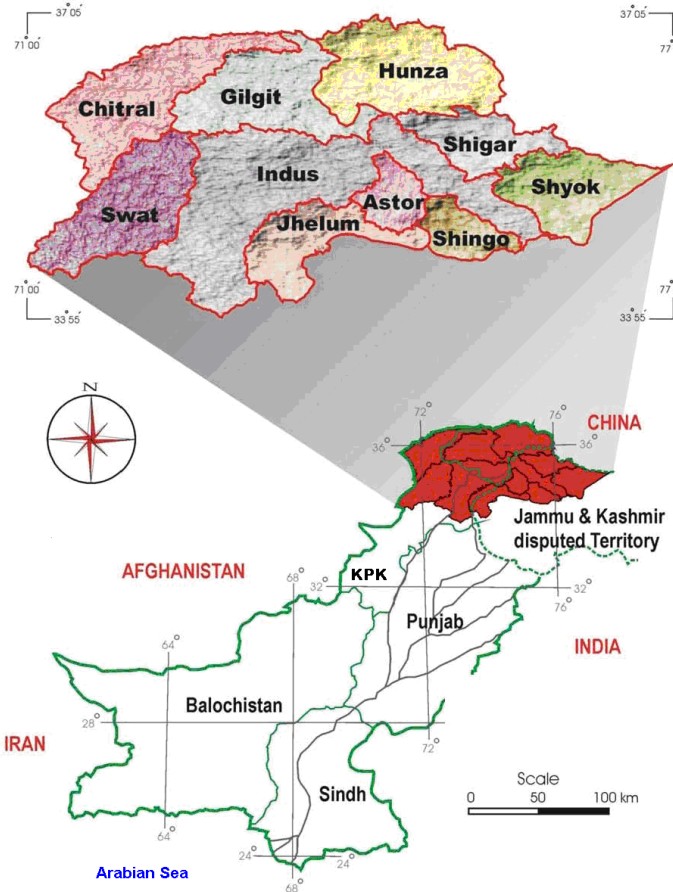

**Figure 1.** Glaciated River basins of northern Pakistan.

2.3.1  **Hunza River Basin**
The Hunza River basin actually forms the sub basin of the Gilgit River but due to its
considerable size and importance it is considered as a separate basin. The river drains the
Karakoram Mountains comprising of large glaciated area in the north (Ashraf et al., 2012).
The Karakoram highway linking Pakistan to China passes across this basin. Part of the road
runs along Hunza River and ends near Khunjerab Pass (Geerken and Bräker, 2017). The
tributaries joining the Hunza River are Chabursan, Khunjerab, Ghujerab, and Shunsha River.
The basin comprises of major valleys and hanging glaciers on the high Karakoram Range
(Figure 2). Karimabad, the capital of the Hunza valley, is stretched over miles and miles of
terraced fields and fruit orchards. It offers a panoramic view of the Rakaposhi, Ultar and





Balimo peaks. Gulmit is shining white and deeply crevassed - just as you would expect a
glacier to look. Above this glacier to the left is the jagged line of the Passu and Batura peaks,
seven of which are over 7,500 m. Passu is the setting-off point for climbing expeditions up
the Batura, Passu, Kurk and Lupgar groups of peaks, and for trekking trips up the Shimshal
Valley and Batura Glacier (Singh, 2015, Kreutzmann, 2018).

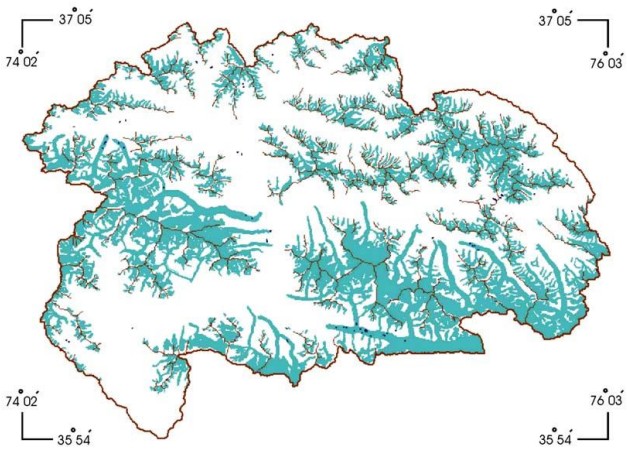

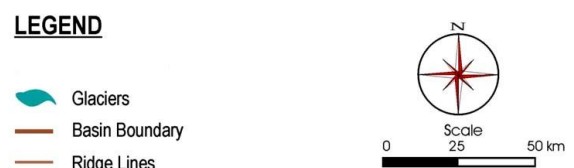

**Figure 2.** The glacier distribution in Hunza River basin.

2.3.2   **Shyok River Basin**
The Shyok River is bounded with Jammu and Kashmir disputed Territory in south,
China in northeast and Shigar and Indus River basins in the west. The elevation in the basin
varies from more than 2,500 masl to more than 7,700 masl (ul Hassan et al., 2018). There are
372 glaciers which contribute to a vast glacier area of about 3,548 Km$^2$ (Figure 3). Though
the Valley glaciers are only 14% of the total number; they contribute more than 82% to the
glacier area. This high contribution is mainly due to larger area of the individual glaciers.
Some of the important valley glaciers include Siachen, Kondus, Bilafond, Chogolisa,
Ghandogoro and Masherbrum (Wolovick, 2016). The glacier area of the basin contributes to
about 892 km$^3$ of the total ice reserves of the basin. Again, the major source of this huge ice
reserve is the valley glaciers which contribute more than 94%. Aspect wise the basin has
been divided into various ordinal directions. Glaciers are oriented towards NE (29%), NW
(24%) and E (18%) but are absent on the western aspects. The total area 33470 km$^2$ is
bounded by $75^0$ to $77^0$ E, $34^0$ to $35^0$ N (Ghosh, 2003, Pfeffer et al., 2014).



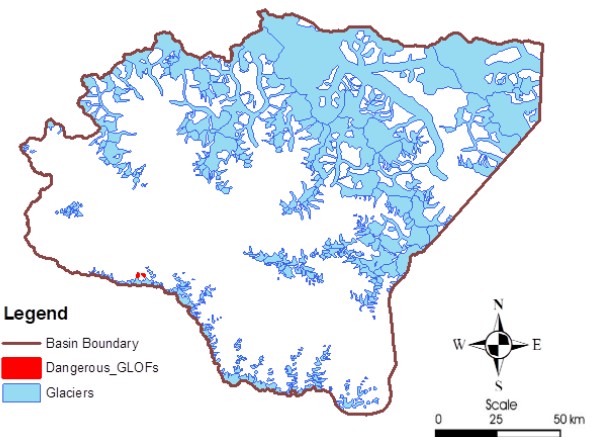


**Figure 3.** The glacier distribution in Shyok River basin.

### 2.4   Dataset

Landsat ETM+ Images of Hunza and Shyok basins, within the substantial time span
from May to September, have been acquired from the USGS (United States Geological
Survey) using the Earth Explorer interface (http://earthexplorer.usgs.gov/), Digital Elevation
Model of Hunza and Shyok basins is used for obtaining the elevation, aspect, and slope of the
glaciers hosting the glacial lakes. ASTER interpolated data at 15m is used for this purpose,
Geomorphologic data of Hunza and Shyok basins of Pakistan acquired from the Geological
Survey of Pakistan.

### 2.5   Methodology

The Study encompasses acquisition of Satellite Images, performance of geospatial
analysis, and identification of GLOFs to assess the glacial hazard-prone areas. By utilizing
height information obtained through DEMs, orientation and slope maps are formed (Nabi et
al., 2018). The Landsat images of different time spans were downloaded and studied in detail
for quality input. Capturing Digital Data of Glacial Lakes from Imagery Landsat images
were used for identification of glacial lakes by applying the Normalized Difference Water
Index (NDWI), taking advantage of the low water reflectance in the NIR band.

$$NDWI = \frac{NIR - Blue}{NIR + Blue}$$

Thereafter, mapping of Glacial Lakes of Shyok basin in contact with glaciers and
upstream of settlements was carried out. In this connection, direct hydrological connection
and lake dam type was determined. The lakes volume was calculated based on surface area.
Finally, the simulation and modeling of potentially dangerous glacial lakes in HEC-RAS was
conducted. The HEC-RAS model (Figure 4) contains several river analysis components for
steady flow water surface profile computations and one and two-dimensional unsteady flow
simulation including velocity and water surface depth analysis. The release of Version 5.0



introduced two-dimensional modeling of flow as well as sediment transfer modeling
capabilities. The program was developed by the US Department of Defense, Army Corps of
Engineers in order to manage the rivers, harbors, and other public works under their
jurisdiction; it has found wide acceptance by many others since its public release in 1995
(Osti and Egashira, 2009).

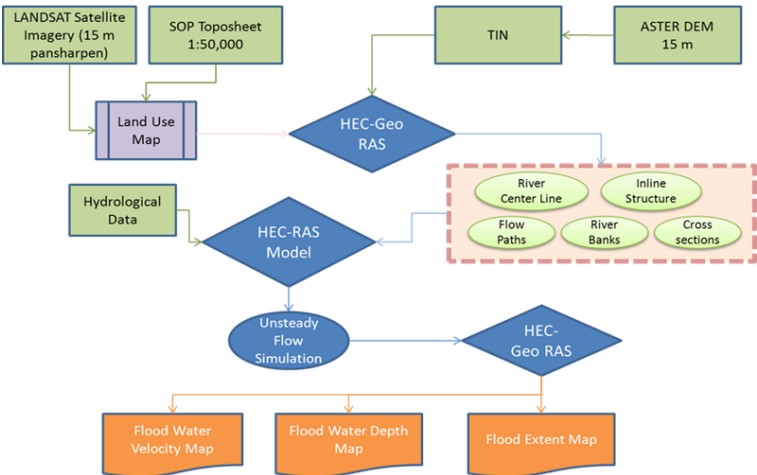


**Figure 4.** HEC-RAS model workflow.


3. **RESULTS AND DISCUSSION**
The potentially dangerous glacial lakes which are concentrated at the headwaters of these
river basins can affect settlements, infrastructure, and agricultural fields situated in the
downstream river valley (Stäubli et al., 2018). Antecedent glaciers and glacial lakes
comprehensive and accurate knowledge are of utmost importance. The ability of
decision-makers on the adaption of risk mitigation measures and reduction in vulnerability
will be enhanced with a detailed digital data repository of glacial lakes and GLOFs
occurrences. This forms the basis for global warming studies and future climate change
research in Pakistan, as the irrigation network is primarily dependent on summer season
snowmelt (Mukhopadhyay and Khan, 2015).
3.1 **GLOFs Hazard Assessment**
By using Landsat-7 images, the study of glaciers and glacial lakes is carried out
coupled with field investigations of potentially dangerous GLOFs. Using remote sensing
satellite images, the monitoring of the glaciers as per created inventories and the impact
assessment of the GLOFs extent is done precisely. The accuracy is achieved with the remote
sensing data and techniques for evaluation of geophysical conditions of the terrain with the
help of satellite images. The ability and precision of the analysis performed is increased with
the multistage approach of field investigation coupled with remote sensing dataset. The
study involving glaciers and GLOFs becomes reliable once visual image interpretation
techniques are integrated with GIS analysis.



For this research, the identification of glaciers and glacial lakes has been done by
utilizing Landsat-7 ETM+ images. Landsat from an altitude of 705 km amsl covers an area of
183 km by 170 km. It is a sun-synchronous orbit imaging after every 16 days and obtains a
synoptic view at an inclination of 98.2 degrees. It carries the ETM+ sensor. The bandwidth
of TM and ETM+ are slightly different ranging from the blue to far infrared wavelength. For
feature identification, Landsat-7 band combinations and indices are utilized. The glacial
lakes can be easily identified in the band combination of RGB (Red-Green-Blue) (Pan-7-6b)
due to their better contrast with the surrounding features. In this FCC, the fresh snow and ice
of the glaciers appear in light to dark red color. In the image of the winter season, the glacial
lakes with a smooth texture and varying gray tone due to their semi-frozen ice surface are
easily identified (Figure 5(a-c)).

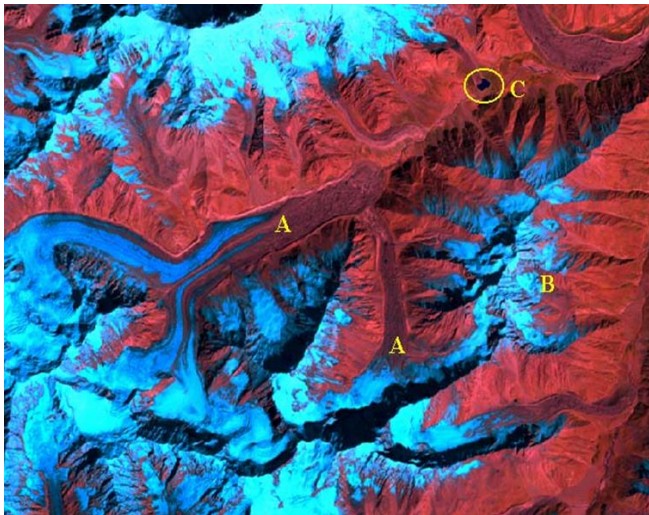

**Figure 5.** Glacier ice covered with debris (a), snow (b) and glacial lake (c).
The identification/monitoring of glaciers and glacial lakes is done with integration of
remote sensing technique and GIS (Geographic Information System) data analysis. It played
a major role in decision making and application of rules of land cover types and features
discrimination in GIS analytical techniques, which enabled better presentation and
perspective views. DEM is utilized to create slope and aspect data sets of the study area.
Even though the glacial lake surfaces are flat and covered by snow, the glaciers ice and snow
ice create slope angles (Gilany and Iqbal, 2016).  Antecedent, decision rules of integrated
GIS analysis is applied, that if the surface texture is smooth and the slope is not pronounced
then such areas are recognized as glacial lakes.
3.2    **GLOFs Analysis of Hunza Basin**
There are 110 glacial lakes covering an area of about 3.22 sq. Km out of which, 47 are
major glacial lakes in this basin. Maximum of these are Supraglacial lakes (20) because in
the basin there are large size glaciers. This type is followed by Valley lakes (17). The other



types of lakes are few in number and therefore contribute very little to the accumulative area
of the major lakes of the basin. In this basin maximum lake area (49%) is contributed by
Valley lakes followed by Supraglacial lakes (28%). Rest of the 23% lake area is collectively
contributed by Moraine dammed, Erosion and Blocked lakes. The largest End Moraine
dammed glacial lake has an area 0.12 sq. Km and is at a distance of 175 m from Passu Glacier
having an area of 62.9 sq. Km, length of 26 km and ice reserves of 10.89 km$^3$. The high relief
and unstable deposits along the valley sides have made the slopes prone to mass movements.
The upper Hunza basin provides an ideal and easily accessible location for the study of
ice-dammed and mass movement-dammed lakes. The largest sized lake in this category has
0.38 sq. Km area and a length of 5000 m. It is oriented towards the North West and is
associated with the Khurdopin glacier. This potentially dangerous glacial lake is hazardous
to the settlement of Shimshal valley (Figure 6a).

### 3.3 HEC-RAS Model Simulation of Glacial Lake Outburst Flood Hazard Risk in Hunza Basin

The HEC-RAS model simulation is utilized for identification of GLOF hazard risk
extents to Shimshal village in Hunza basin. Shimshal village is located in Gojal Tehsil of
Hunza District, in the Gilgit–Baltistan region of Pakistan. It lies at an altitude of 3,100 m
amsl and is the highest settlement in Hunza basin. It is a border village that connects the
Gilgit-Baltistan with China. The total area of Shimshal is approx 3,800 km$^2$ and there are
around 2000 inhabitants with a total of 250 houses. The input parameters for the HEC-RAS
model are listed in table 1.

**Table 1.** HEC-RAS input parameters (Brunner, 2002).

| Input Parameters | Value Assigned | Description |
|---|---|---|
| DEM | 30 m | Digital Elevation Model (ASTER) For slope angle, altitude, and curvature |
| Inlet | Polyline | The area with specific release drawn in shapefile |
| Global Parameters | Volume | Discharge volume (m$^3$/s) |
| | Intake Period | Time interval in mins (1,3 and 30) |
| Energy Slope | 0.1 | Energy slope for distributing flow along boundary condition |
| Domain Area Mesh | Perimeter | Domain area with specific perimeter bound around the area of interest drawn in shapefile |

#### 3.3.1 Khurdopin Glacial lake Inlet and Shimshal Village Domain Area

First, the domain area surrounding the flow accumulation of stream flowing out of
potentially dangerous glacial lake is drawn. The habitat of Shimshal village is included in the
domain area to ascertain the damage extents to the settlements. The two-dimensional domain
area is assigned the pixel value of 15x15m. Inlet to the domain area is drawn at the outflow of
potentially dangerous Khurdopin glacial lake (Figure 6 b).





The peak seasonal discharge from potentially dangerous lake of Khurdopin glacier is
calculated using the empirical formula of peak discharge (Costa, 1988).
$$Qmax = 113(Vo \times 10^{-6})^{0.64} \qquad (1)$$
Where,
$Qmax$ = Peak Discharge (m³/s)
$Vo$ = Volume (m³)
Basing on the parameters (Table 2) of the potentially dangerous khurdopin glacial lake
(Figure 6c), the peak seasonal discharge flow is calculated as 3500 m³/s, which can generate
low flood damage extent to the settlements of Shimshal village (Table 2). The peak
simulated scenario-1 discharge flow is calculated as 5000 m³/s, which can generate moderate
flood damage extent the peak simulated scenario-2 discharge flow is calculated as 7000 m³/s,
which can generate high flood damage extent to Shimshal village (Table 2).
**Table 2.** Parameters of potentially dangerous glacial lake to Shimshal village.

| Parameters | Peak Seasonal Value | Simulated Scenario-1 Value | Simulated Scenario-2 Value |
|---|---|---|---|
| Length | 5000 m | 5500 m | 6000 m |
| Depth | 100 m | 150 m | 200 m |
| Aspect | NW | NW | NW |
| Area | 2464780 (m)² | 3081236 (m)² | 3652018 (m)² |
| Volume | 246478000 (m)³ | 308123600 (m)³ | 365201800 (m)³ |
| Discharge | 3500 m³/s | 5000 m³/s | 7000 m³/s |

3.3.2   **HEC-RAS Simulated Hydrograph Max Depth, Max Velocity, and Max Water**
**Surface Elevation at X–Sec of Shimshal Village**
Basing on a peak seasonal discharge of 3500 m³/s and data input interval of 10 mins,
the two-dimensional hydrograph profiles are generated from HEC-RAS model. The max
depth of flow hydrograph at Shimshal village x-sec is calculated as 40 m, which has
generated a low flood to the settlements (Figure 6d). The max velocity of flow is calculated
as 7 m/s (Figure 6e). The reference surface elevation is 3070 m amsl and the max water
surface elevation at Shimshal village x-sec is 3110 m amsl (Figure 6f).

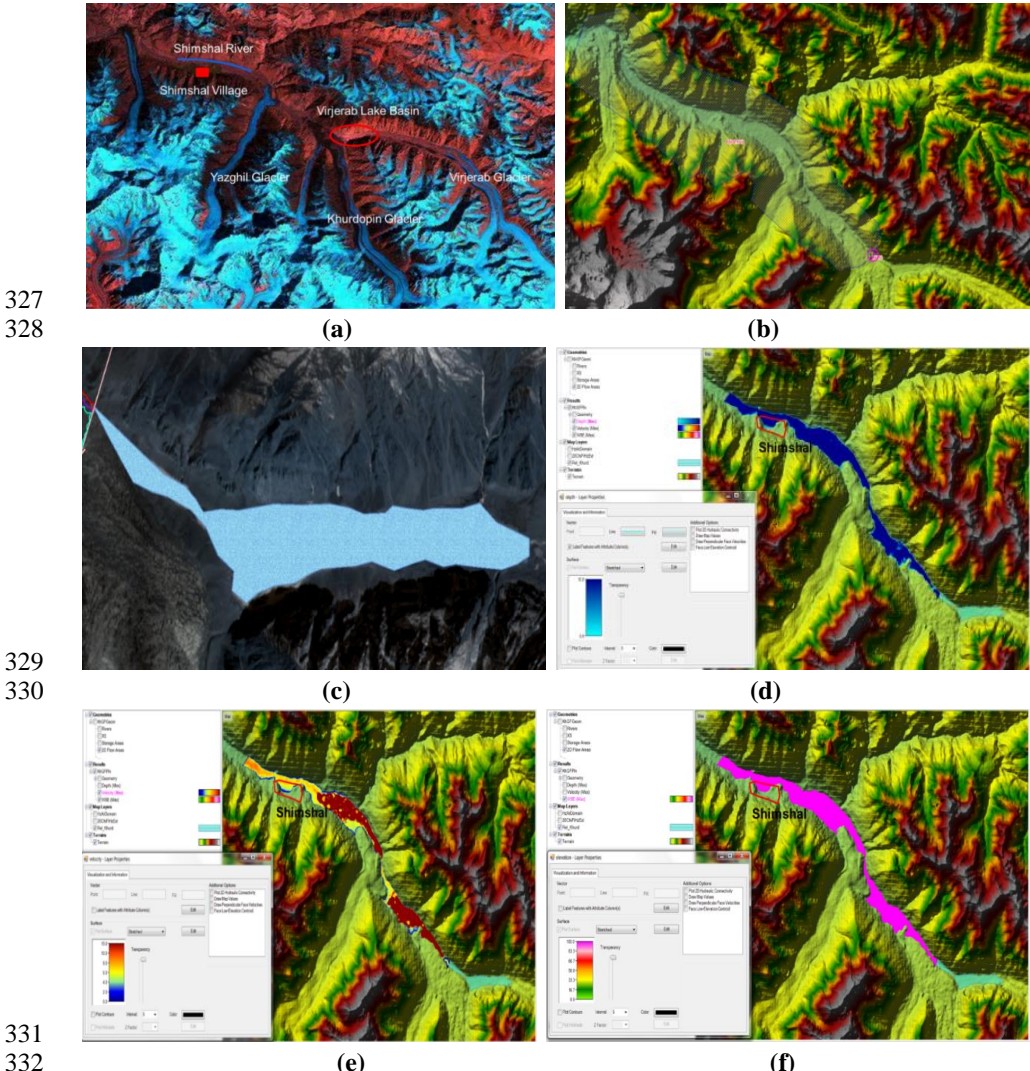

(a) (b)
(c) (d)
(e) (f)

**Figure 6.** (a) General area showing glacial lake to Shimshal village, (b) Glacial lake inlet and
Shimshal village domain area, (c) Potentially dangerous lake of seasonal low discharge, (d)
Depth profile at x–sec of Shimshal village, (e) Velocity profile at x–sec of Shimshal village,
(f) Water surface elevation profile at x–sec of Shimshal village.






### 3.3.3 HEC-RAS Output Parameters at Shimshal Village X-SEC (Low, Moderate and High Discharge)


Basing on the simulated 2D hydrograph profiles generated from HEC-RAS model,
the output parameter values of potentially dangerous Khurdopin glacial lake at Shimshal
village x-sec generating low, moderate and high discharge are as shown in table 3.
**Table 3.** HEC-RAS outputs at Shimshal village x-sec (low, moderate and high discharge).

| Output Parameters | Peak Seasonal (Low Discharge) | Simulated Scenario-1 (Moderate Discharge) | Simulated Scenario-2 (High Discharge) | Description |
|---|---|---|---|---|
| Flow Height | 40 m | 50 m | 70 m | Flow height obtained during the course of GLOF |
| Velocity Generated | 7 m/s | 8 m/s | 10 m/s | The velocity of turbulent water of GLOF |
| WSE | 3110 m | 3120 m | 3140 m | Water Surface Elevation |
| Damage Extent | 950 m | 1100 m | 1250 m | The extent of damage by glacial lake outburst flood |

Keeping in view the seasonal growth of potentially dangerous glacial lake in Hunza
basin and simulated scenarios, the low discharge of 3500 $m^3$/s from Khurdopin glacial lake
can affect 40% of the Shimshal village habitat, the moderate discharge of 5000 $m^3$/s from
Khurdopin glacial lake can affect 60% of the Shimshal village habitat and the high discharge
of 7000 $m^3$/s from Khurdopin glacial lake can affect 80% of the Shimshal village habitat as
shown below in figure 7(a-c).





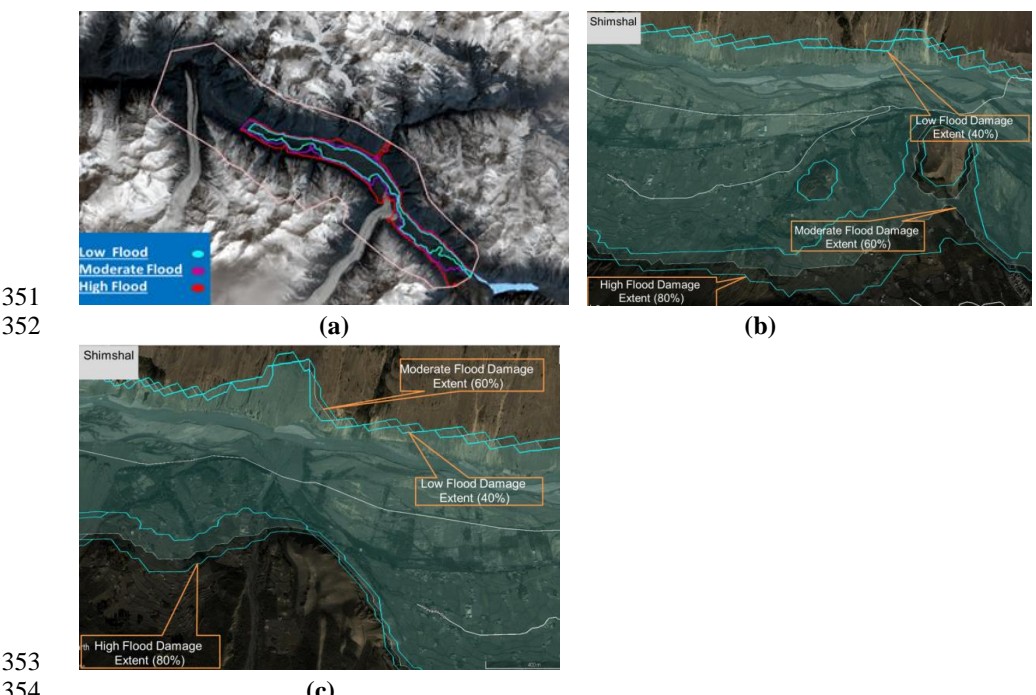

**(a)**
**(c)**

**(b)**

**Figure 7. (a-c)** GLOF damage extents of Shimshal village habitat.



### 3.4 GLOFs Analysis of Shyok Basin


This basin comprises of a glaciated area of about 3,548 km$^2$ out of a total area of 10,235
km$^2$. The distribution of different types of the glacial lakes shows the pronounced hazard of
GLOFs. The Shyok basin's 2.7 sq. km area is covered with 66 glacial lakes. In this basin,
most of the glaciers are concentrated in the north-eastern part while the glacial lakes are
scattered over the south-western part (Hewitt, 1998). Most of the lakes (39%) are of erosion
type covering an area of approximately 0.5 sq. Km. Though the End Moraine and Valley
lakes are only twelve and eight in number, respectively, but they contribute about 40% and
30% of the lake area respectively. The Eight Valley lakes in the basin contribute more than
29% of the lake area. Most of these lakes are at varying distances from their associated
glaciers. Half of these eight lakes are closed type lakes. The largest Valley glacial lake is
associated with Siachen glacier and is oriented towards the north. It has 0.27 sq. km area and
a length of 670 m. The End Moraine lakes are twelve which contribute about 40% to the lake
area. The largest sized lake in this category has 0.21 sq. km area and a length of 800 meters.
It is oriented towards the north and is associated with the Siachen glacier. Potentially
dangerous lakes of this basin are hazardous to the settlements of Barah village (Figure 8).

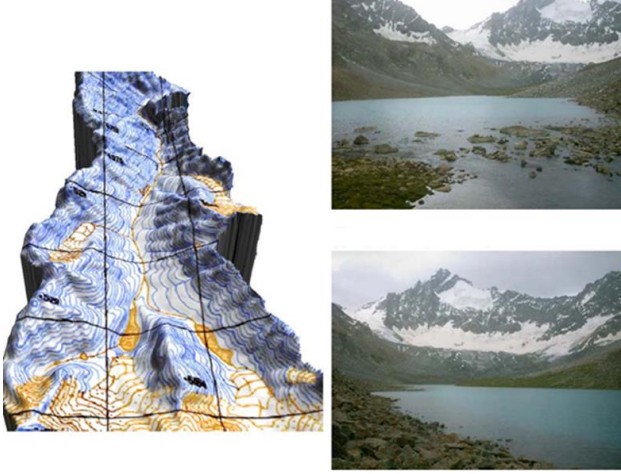

**Figure 8.** Potentially dangerous glacial lakes to Barah village.

### 3.5 HEC-RAS Model Simulation of Glacial Lake Outburst Flood Hazard Risk in Shyok Basin


The HEC-RAS model simulation is utilized for identification of GLOF hazard risk
extents to village Barah in Shyok basin. The input parameters for the HEC-RAS model are
listed in table 1.

### 3.5.1 Potentially Dangerous Glacial Lakes Inlets and Barah Village Domain Area


First, the domain area surrounding the flow accumulation of stream flowing out of
both potentially dangerous glacial lakes are drawn. The habitat of Barah village is included
in the domain area to ascertain the damage extents to the settlements. The two-dimensional





domain area is assigned the pixel value of 15x15m. Inlets to the domain area are drawn at the
outflow of potentially dangerous glacial lakes (Figure 9a). The expected peak seasonal
discharge from potentially dangerous lakes of Barah village is calculated using the empirical
formula of peak discharge (equation (1)).
Basing on the parameters of the potentially dangerous glacial lakes (Figure 9b), the
peak seasonal discharge flow is calculated as 100 m$^3$/s (Table 4), which can generate low
flood damage extents to the settlements of Barah village.
**Table 4.** Parameters of potentially dangerous glacial lake to Barah village (peak seasonal).

| Parameters | Value (Left Lake) | Value (Right Lake) |
|---|---|---|
| Length | 1300 m | 1400 m |
| Depth | 4 m | 3 m |
| Aspect | N | NW |
| Area | 450,368.15 (m)$^2$ | 517,112.16 (m)$^2$ |
| Volume | 1,801472.60 (m)$^3$ | 1,552236.48 (m)$^3$ |
| Latitude | 35°05'53.78"N | 35°05'50.47"N |
| Longitude | 76°14'15.55"E | 76°15'19.36"E |

3.5.2  **HEC-RAS Simulated Hydrograph Max Depth, Max Velocity and Max Water**
**Surface Elevation at X–Sec of Barah Village**
Basing on a peak seasonal discharge of 100 m$^3$/s and data input interval of 10 mins,
the two-dimensional hydrograph profiles are generated from HEC-RAS model. The max
depth of flow hydrograph at Barah village x-sec is calculated as 25 m, which has generated a
low flood to the settlements (Figure 9c). The max velocity of flow is calculated as 5 m/s
(Figure 9d). The reference surface elevation is 2560 m amsl and the max water surface
elevation at Barah village x-sec is 2585 m amsl (Figure 9e).





(a)

(b)

(c)

(d)

(e)

**Figure 9.** (a) Potentially dangerous glacial lakes to Barah village, (b) Glacial lakes inlets and Barah village domain area, (c) Depth profile at x–sec of Barah village, (d) Velocity profile at x–sec of Barah village, (e) Water surface elevation profile at x–sec of Barah village.





### 3.5.3 HEC-RAS Output Parameters at Barah Village X-SEC (Low, Moderate and High Discharge)

Basing on the simulated two-dimensional hydrograph profile generated from HEC-RAS model, the output parameters values of potentially dangerous glacial lakes having peak seasonal discharge of 100 m³/s are obtained as shown in table 5.

**Table 5:** HEC-RAS output parameters at Barah village x-sec (low discharge of 100 m³/s).

| Output Parameters | Peak Seasonal (Low Discharge) | Description |
|---|---|---|
| Flow Height | 25 m | Flow height obtained during the course of GLOF |
| Velocity Generated | 5 m/s | The velocity of turbulent water of GLOF |
| WSE | 2585 m | Water Surface Elevation |
| Damage Extent | 500 m | The extent of damage by glacial lake outburst flood |

The peak simulated scenarios discharge from potentially dangerous lakes of Barah village is calculated using the empirical formula of peak discharge (equation (1)).

Basing on the parameters of the potentially dangerous glacial lakes, the peak simulated scenario-1 discharge flow is calculated as 300 m³/s (Table 6), which can generate moderate flood damage extend to Barah village. The peak simulated scenario-2 discharge flow is calculated as 500 m³/s (Table 6), which can generate high flood damage extents to Barah village.

**Table 6.** Parameters of potentially dangerous glacial lake to Barah village (simulated scenarios).

| Parameters | Simulated Scenario-1 Value | | Simulated Scenario-2 Value | |
|---|---|---|---|---|
| | Left Lake | Right Lake | Left Lake | Right Lake |
| Length | 1400 m | 1500 m | 1500 m | 1600 m |
| Depth | 8 m | 7 m | 14 m | 12 m |
| Aspect | N | NW | N | NW |
| Area | 610,368.15 (m²) | 717,112.16 (m²) | 710,368.15 (m²) | 917,112.16 (m²) |
| Volume | 4,882945.20 (m³) | 4,597978.04 (m³) | 9,945154.10 (m³) | 10,214228.38 (m³) |

Basing on the simulated two-dimensional hydrograph profile generated from HEC-RAS model, the output parameters values of potentially dangerous glacial lakes having peak simulated scenario-1 discharge of 300 m³/s and peak simulated scenario-2 discharge of 500 m³/s are obtained as shown in table 7.






**Table 7.** HEC-RAS output parameters at Barah X-SEC (moderate and high discharge).

| Output Parameters | Simulated Scenario-1 (Moderate Discharge) | Simulated Scenario-2 (High Discharge) | Description |
|---|---|---|---|
| Flow Height | 30 m | 35 m | Flow height obtained during the course of GLOF |
| Velocity Generated | 6 m/s | 7 m/s | The velocity of turbulent water of GLOF |
| WSE | 2590 m | 2595 m | Water Surface Elevation |
| Damage Extent | 650 m | 700 m | The extent of damage by glacial lake outburst flood |

**3.6 HEC-RAS Model Simulated GLOF Hazard Extents at Barah Village X-Sec**
Keeping in view the seasonal growth of potentially dangerous glacial lakes of Shyok
basin and simulated scenarios, the low discharge of 100 m$^3$/s from both lakes can affect 20%
of the Barah village habitat, the moderate discharge of 300 m$^3$/s from both lakes can affect
30% of the Barah village habitat and the high discharge of 500 m$^3$/s from both lakes can affect
40% of the Barah village habitat as shown below (Figure 10).

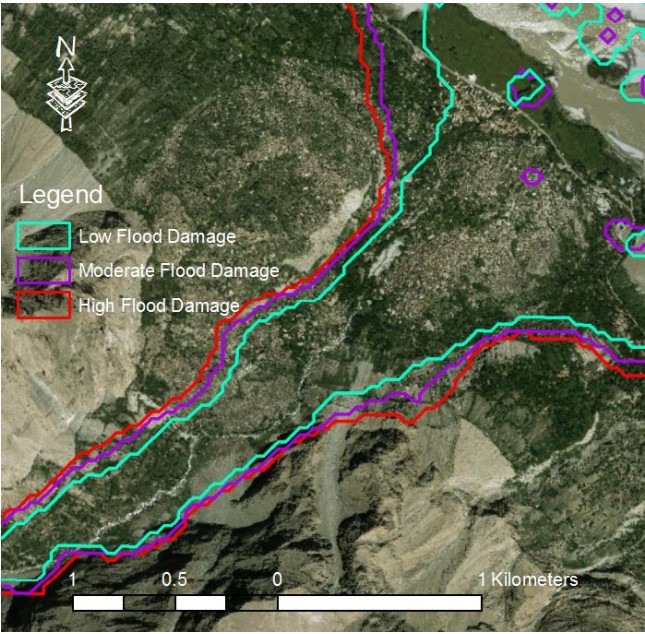

**Figure 10**. GLOF damage extents of Barah village habitat.






4. **CONCLUSION**

The GLOFs play a vital role in sedimentation and erosion in UIB of Pakistan. Their significance cannot be denied especially which lies in very exceptional risk to the infrastructure and human installations. The historically recorded floods gain height well beyond peak discharge estimated values for the seasonal precipitation. The erosion capacity and competence are immensely enhanced by the active dynamic character of the GLOFs. In the context of erosion in these valleys and the sedimentation of the reservoirs in the downstream area, the vital importance is of the GLOFs happening. The passage of this dam burst involving turbulent floods has contributed in huge numbers of landslides which have occurred in valley sides and on the terraces of Hunza and Shyok basins. Keeping in view the seasonal growth of potentially dangerous Khurdopin glacial lake of Hunza basin and simulated scenarios, the low discharge of 3500 $m^3$/s from glacial lake can affect 40% of the Shimshal village habitat, the moderate discharge of 5000 $m^3$/s from glacial lake can affect 60% of the Shimshal village habitat and the high discharge of 7000 $m^3$/s from glacial lake can affect 80% of the Shimshal village habitat. Keeping in view the seasonal growth of potentially dangerous glacial lakes of Shyok basin and simulated scenarios, the low discharge of 100 $m^3$/s from both lakes can affect 20% of the Barah village habitat, the moderate discharge of 300 $m^3$/s from both lakes can affect 30% of the Barah village habitat and the high discharge of 500 $m^3$/s from both lakes can affect 40% of the Barah village habitat. The Shyok basin and Hunza basin are prone to glacial lake outburst floods hazard based on the proximity of glacial lake with respect to infrastructure, geomorphology of underneath surface, geo-cover of the vicinity, crevasses, ice melt, and anthropogenic activities. Therefore, continuous monitoring through physical gauge stations and satellite images is very vital of the streams nearing settlements of Shyok and Hunza basin. Knowing the extent of damages beforehand can help in mitigating the impact of GLOF surges. Antecedent, the most vital mitigation step to reduce flood risk is to gradually reduce the volume of the glacial lake to decrease the dynamic peak surge of glacial lakes containing a huge volume of water. In order to protect the infrastructure in downstream areas against the destructive/dynamic forces of surging GLOFs, pre-disaster mitigation measures must be taken. An early warning and monitoring system should be placed in advance in order to safeguard against such catastrophic events. While choosing the appropriate method or starting any mitigation measure, precise evaluation involving detailed analysis studies of lakes, mother glaciers, surrounding conditions, and damming materials are the foremost requirements. The measures adopted must be such that those must not increase the risk of a GLOF event during or after the placement of mitigation measures. At different stages of the mitigation process i.e., during or after, the onsite monitoring gadgets at the mother glaciers, the lake, the dam, and the surroundings are very vital.

**Acknowledgements**

We are highly grateful to Almighty Allah, the most beneficent, the most merciful, for giving us strength, courage and resources to complete this research. We are sincerely obliged to IGIS-NUST for providing us platform of a knowledge base during the study period. We are very thankful to SUPARCO for cooperation and help in this research work. We acknowledge PMD (Pakistan) for the provision of data support for this paper.



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
