# Peer review of "Geospatial Analysis and Simulation of Glacial Lake Outburst Flood Hazard in Hunza and Shyok Basins of Upper Indus Basin"

_The Cryosphere, 2019_

## Referee Comment (RC1) · Anonymous Referee #1 · 3 Mar 2020

Gilany et al. present an outburst flood modelling study for three glacial lakes in Pakistan. The study is interesting but lacking key details required for its justification. For example, it is not clear why these lakes are 'potentially dangerous' as referred to throughout the study. It should be detailed why these lakes constitute a hazard. The manuscript requires significant English language and grammar editing. The introduction requires significant restructuring, editing, and addition of relevant literature. It is repetitive in several places and does not proceed logically or provide a clear context for the study. Rather it jumps between discussions of glaciers in Pakistan, global glacier retreat, surging glaciers, glacial lakes, and Himalayan glaciers. There are numerous statements throughout the manuscript containing details that are not supported with citations. The study requires significant attention in order to properly ground it in existing literature. The flow modelling procedure is lacking basic details on how it was carried out and how parameters were selected, which means it is difficult to review the results. Similarly, several flood modelling figures are lacking readable legends. It is also not clear from the figures that the model domain was large enough to avoid flood modelling errors, or that the DEM used is suitable, or what the associated uncertainties are in the modelling approach and simulated flood extents. I suspect that the manuscript may be suitable for publication in The Cryosphere after significant revisions, but it may fit better in another journal.

Specific comments: L10 'anthropogentic factors'.

L18 Why is the water 'turbulent'? Do you mean the ASTER GDEM? If so cite the appropriate version and change in the later text.

L43 'Indian subcontinent'.

L44 This is not an appropriate reference relating to downstream water supply. See Immerzeel et al. (2010), Quincey et al. (2018), Immerzeel et al. (2020).

L48 Add a citation for 'many disastrous glacial lakes'. 'The damming' of what?.

L51-52 This is incorrect.

L53-55 This needs a citation unless it is related to Che et al, in which case make it clear.

L71-73 This is repeating what has already appeared in the introduction.

L79-80 Also repetition.

L80-81 This sentence on Baltoro is out of place.

L84-85 Surges are not necessarily coupled with glacial lakes.

L95-97 A lake does not form in all cases.

L98 Vital for the stability of what, the lake?

L99 What is a 'heavy accumulation zone'?

L109 'quite similar in length'?

L111-112 Provide a citation. Is this on average? I'm not sure it is really relevant.

L133 Superscript '2'

L152 Provide a citation.

L166 I think the grey shading between the panels is unnecessary. What are the brown coloured lines within the green polygons? Please show the location of all lakes modelled in this figure.

L176-178 I'm not sure that 'panoramic views' are relevant to the study, nor the following sentence.

L183 Specify the source of the glacier outlines. It would be useful to add an elevation model or river network to show the basin topography and flow direction.

L189 What are the other glaciers in addition to 'valley glaciers'?

L197 Use degree symbols.

L199 This figure could be added as a panel alongside Figure 2.

L205 Is this a version of the ASTER GDEM or a DEM you have generated using ASTER data? How were the data interpolated and why?

L214-215 Provide an NDWI citation.

L219 Specify the HECRAS version you used and whether it was a 1D or 2D model. Refer to other studies that have used HEC-RAS to model GLOFs, including limitations such as consideration of sediment loading and bank erosion.

L233-234 The meaning of this is unclear.

L256 What is 6b?

L270 What are the specific thresholds used for these decision rules?

L279 What is 'erosion and blocked lakes'?

L280 Use km2 for consistency.

L285 '5 km'.

L297 Earlier it was stated that you used a 15 m DEM. How was the energy slope derived?

L305 Detail why this empirical relationship is suitable for application to the lakes in your study.

L316 Quote fewer significant digits. Does depth refer to a max/mean depth?

L319 What is the 'data input interval'? Is this the complete duration of the simulation, or time to peak discharge?

L334 Why is this a potentially dangerous lake? The text is not visible on panels d-f. Why does it look like the model formed a large lake on (d)?

L343 Why is the damage extent in metres. Shouldn't this be an area?

L347 What was the source of information for the village areas?

L355 The village/buildings should be marked for clarity. Add a scale and north arrow.

L374 The 3D model on the left needs further information adding (legend, lake location, markers etc).

L393 These lake depth values are very small. What is the source of the data?

L402 The panels need legends.

L411 Specify what you mean by 'damage extent' and other 'output parameters'. Are

these max values, or an average?

References Immerzeel, W.W., Lutz, A.F., Andrade, M., Bahl, A., Biemans, H., Bolch, T., Hyde, S., Brumby, S., Davies, B.J., Elmore, A.C., Emmer, A., Feng, M., Fernández, A., Haritashya, U., Kargel, J.S., Koppes, M., Kraaijenbrink, P.D.A., Kulkarni, A.V., Mayewski, P.A., Nepal, S., Pacheco, P., Painter, T.H., Pellicciotti, F., Rajaram, H., Rupper, S., Sinisalo, A., Shrestha, A.B., Viviroli, D., Wada, Y., Xiao, C., Yao, T., and Baillie, J.E.M. (2020). Importance and vulnerability of the world's water towers. Nature 577, 364-369. Immerzeel, W.W., Van Beek, L.P.H., and Bierkens, M.F.P. (2010). Climate change will affect the asian water towers. Science 328, 1382-1385. Quincey, D., Klaar, M., Haines, D., Lovett, J., Pariyar, B., Gurung, G., Brown, L., Watson, C., England, M., and Evans, B. (2018). The changing water cycle: the need for an integrated assessment of the resilience to changes in water supply in High-Mountain Asia. Wiley Interdisciplinary Reviews: Water 5, e1258-n/a.

---

## Author Comment (AC1) · 4 Mar 2020

Thanks for rating the study as interesting and for your constructive suggestions concerning key details required. We presented an outburst flood modelling study for three glacial lakes in Pakistan. These lakes are referred as 'potentially dangerous' because of their potential outburst and being hazardous to the downstream settlements. These lakes constitute a hazard owing to their substantial seasonal growth. The requisite English language and grammar editing of manuscript will be done in the final revised version. The requisite restructuring and editing of introduction including addition of more literature will be incorporated in the final revised version of the manuscript. The

repetitions at places will be done away with for its logical proceed to provide more clear context for the study. The observations regarding results will be incorporated in the final revised version of the manuscript. The model domain was large enough to avoid flood modelling errors and the DEM used of 15 m resolution was suitable for simulated flood extents. Moreover, the specific comments will be addressed in the final revised version of the manuscript.

---

## Referee Comment (RC2) · Anonymous Referee #2 · 8 Mar 2020

The manuscript presents glacial lake hazard assessment in two river basins in the Upper Indus Basin. GLOFs have been a serious concern in the mountainous regions over the last two decades. Especially in the Himalaya, the climate-driven glacier retreat is contributing to the growth of the high altitude lakes. These lakes may present a great threat to the downstream regions. Ascertaining the GLOF hazard in the Himalaya is extremely important and relevant in the current scenario of climate change. This paper has potential in this regard, however, the present study does not succeed incomprehensively addressing the issue. There are many important issues that need to be addressed and suggestions which need to be incorporated. The detailed general and section-wise remarks on the manuscript have been outlined below. This study

mostly concentrates on GLOF assessment using remote sensing and modeling approaches (HEC RAS). I am not sure if the study fits well in the scope of the journal "The Cryosphere", however, such studies are more suitable in NHESS or Natural Hazards.

General comments:

1. The abstract is too long and general. It lacks in representing the importance of the given study. 2. The methods are not clearly outlined in the manuscript. It is poorly organized and contents of different sections overlap. 3. The results produced in the study are not sufficient to support the interpretations and conclusions. The discussion section lacks a comparative analysis, the results do not show any quantitative comparison with other studies in the region. 4. The English needs improvement in the entire manuscript.

Section-wise Comments:

Introduction:

The introduction lacks the latest literature on GLOF modeling studies in the Himalaya. (see the references below). I would suggest to summerise the structure of the paper in a few lines towards the end of the section.

Materials and methods

1. The sections (2.1, 2.2, 2.3) can be combined together into a common section as "Study regions and climate". 2. Sections ( 2.4 and 2.5) can be combined as "Data and Methods" 3. The details of the remote sensing datasets used in the study are missing. I do not understand why is ASTER DEM interpolated to 15 m as it has an actual resolution of 30m. 4. Area-based scaling has been used to calculate the volume of the lakes. However, it is not clear which empirical relation has been adopted for the calculations. ( Refer to Huggel et al., 2004; Cook and Quincey,2015; Fujita et al., 2013). As this is the most crucial factor in GLOF hazard evaluation, it should be discussed. 5. The

flow chart and the methodology doesn't explain about the breach hydrograph. How is it obtained? What are the breach parameters? Mechanism of failure?. The methodology sections need to be revised and rewritten giving more emphasis on the GLOF parameters and flood hydrographs. There is a lot of overlap between the methods and results in the presented manuscript.

Results and Discussion:

This section is too short and vaguely written. It does not provide all the required details of the results obtained. This section should be thoroughly rewritten. 1. In section 3.1 – It is more of a methodology than results. The number of lakes and hazard evaluation criteria for selecting the specific lakes for this study remains unclear. 2. In Section 3.2 the text mostly explains about the classification of the lakes and does not fit well in the section as the section reads as "GLOF analysis of HUza basin" 3. The hazard criteria adopted in the study does not explain the thresholds used for dam geometry, freeboard, and potential lake impacts.

Section 3.3 1. The results and methods are not well separated here. The input parameters of the hydrodynamic model fits well in the method section than the results. 2. Section 3.3.1 is not clear how flood volumes were considered for the different GLOF scenarios. 3. The potential flood hydrographs produced in scenario modeling is not shown. The initial breach hydrograph is most crucial in any GLOF analysis as it determines the flood hydraulics downstream as the GLOF propagates along the flow channel. This section needs to be rewritten showing results of the breach parameters and flood hydrographs. 4. In section 3.3.2 and 3.3.3, the routing parameters are not clear, there has been no mention of the surface roughness along the flow channel. 5. The boundary condition (upstream and downstream) for routing the potential GLOF event remains unclear. 6. Are the flow depths/ velocities representing the mean value along the crosssection or just at a specific point? 7. There has been no mention of the flood wave arrival timings at specific sites along the flow channel. 8. The above comments apply for sections 3.4 to 3.6

Figures:

All figures lack in proper resolution. I recommend exporting the figures with a minimum resolution of 300 dpi for more clarity. Figures 2 and 3 can be combined. The drainage of the basin and the location of the potentially inundated settlements are not shown in the figures. Figure 5 can be removed. Figure 6-The figure lacks locational information. The legends remain unclear. The figure can be better represented using other GIS platforms instead of RAS MAPPER. Figure 5-The figure lacks locational information and legends. Figure 9-see comments for figure 6.

Overall, the presented GLOF hazard assessment requires significant reworking. I hope incorporating the above-mentioned comments can be helpful to improve the presented study. I, therefore, recommend a major revision.

References:

Cook, S.J.; Quincey, D.J. Estimating the volume of Alpine glacial lakes. Earth Surf. Dyn. Discuss. 2015, 3, 559–575.

Worni et al., 2013. Glacial lakes in the Indian Himalayas—From an area-wide glacial lake inventory to on-site and modeling based risk assessment of critical glacial lakes. Sci. Total Environ. 2013, 468,

Huggel, C.; Haeberli, W.; Kääb, A.; Bieri, D.; Richardson, S. An assessment procedure for glacial hazards in the Swiss Alps. Can. Geotech. J. 2004, 41, 1068–1083.

Sattar, A., Goswami, A., Kulkarni, A., & Emmer, A. (2020). Lake Evolution, Hydrodynamic Outburst Flood Modeling and Sensitivity Analysis in the Central Himalaya: A Case Study. Water, 12(1), 237.

Fujita, K., Sakai, A., Takenaka, S., Nuimura, T., Surazakov, A. B., Sawagaki, T., & Yamanokuchi, T. (2013). Potential flood volume of Himalayan glacial lakes. Natural Hazards & Earth System Sciences, 13(7).

[Figure]

Sattar, A.; Goswami, A.; Kulkarni, A.V. Application of 1D and 2D hydrodynamic modeling to study glacial lake outburst flood (GLOF) and its impact on a hydropower station in Central Himalaya. Nat. Hazards 2019, 97, 535–553.

Froehlich, D.C. Peak ouTflow from breached embankment dam. J. Water Resour. Plan. Manag. 1995, 121, 90–97.

Sattar, A.; Goswami, A.; Kulkarni, A.V. Hydrodynamic moraine-breach modeling and outburst flood routing—A hazard assessment of the South Lhonak lake, Sikkim. Sci. Total Environ. 2019, 668, 362–378.
* * *

---

## Author Comment (AC2) · 9 Mar 2020

Thanks for your comment as, "this paper has potential in this regard", and for your constructive suggestions concerning key details required. We are grateful for your constructive input on the manuscript. The feedback has helped us to improve the clarity and structure of the manuscript. We presented an outburst flood modelling study for two River basins in Pakistan. Surely the GLOF hazard in the Himalaya is extremely important and relevant in the current scenario of climate change. The lakes being discussed in the study present a great threat to the downstream settlements.

General comments and answers:

[Figure]

1. The abstract is too long and general. It lacks in representing the importance of the given study.

The abstract has been shortened and the importance of the given study has been represented in the revised version of the manuscript.

2. The methods are not clearly outlined in the manuscript. It is poorly organized and contents of different sections overlap.

The revised version of the manuscript has been better organized and methods are clearly outlined.

3. The results produced in the study are not sufficient to support the interpretations and conclusions. The discussion section lacks a comparative analysis, the results do not show any quantitative comparison with other studies in the region.

The discussion section has been augmented with comparative analysis and the results produced support the conclusions.

4. The English needs improvement in the entire manuscript.

The requisite English language and grammar editing of manuscript has been done in the final revised version.

Section-wise Comments and answers

Introduction:

The introduction lacks the latest literature on GLOF modeling studies in the Himalaya. (see the references below). I would suggest summarizing the structure of the paper in a few lines towards the end of the section.

The requisite restructuring and editing of introduction including addition of more literature has been incorporated in the final revised version of the manuscript.

Materials and methods:

[Figure]

1. The sections (2.1, 2.2, and 2.3) can be combined together into a common section as "Study regions and climate".

It has been re-organized.

2. Sections (2.4 and 2.5) can be combined as "Data and Methods"

For clarity sack these sections have been kept separate.

3. The details of the remote sensing data sets used in the study are missing. I do not understand why ASTER DEM is interpolated to 15m as it has an actual resolution of 30m.

The ASTER GDEM has been interpolated to 15m resolution to match the domain area mesh of 15m x 15m. The details of the remote sensing data sets utilized in the study are given in section 2.4.

4. Area-based scaling has been used to calculate the volume of the lakes. However, it is not clear which empirical relation has been adopted for the calculations. ( Refer to Huggel et al., 2004; Cook and Quincey,2015; Fujita et al., 2013). As this is the most crucial factor in GLOF hazard evaluation, it should be discussed.

The volume calculations of the lakes are based on area and depth (Refer table 2 of the study). Volume=Area*Depth

5. The flow chart and the methodology doesn't explain about the breach hydro-graph. How is it obtained? What are the breach parameters? Mechanism of failure?. The methodology sections need to be revised and rewritten giving more emphasis on the GLOF parameters and flood hydro-graphs. There is a lot of overlap between the methods and results in the presented manuscript.

The methodology section has been revised with more emphasis on the parameters of flood hydro-graphs. The overlap has been removed.

Results and Discussion:

This section is too short and vaguely written. It does not provide all the required details of the results obtained. This section should be thoroughly rewritten.

1. In section 3.1 – It is more of a methodology than results. The number of lakes and hazard evaluation criteria for selecting the specific lakes for this study remains unclear.

The identification of glacial lakes and hazard potential criteria for selecting the specific lakes for this study has been incorporated.

2. In Section 3.2 the text mostly explains about the classification of the lakes and does not fit well in the section as the section reads as "GLOF analysis of Hunza basin"

The section has been revised.

3. The hazard criteria adopted in the study does not explain the thresholds used for dam geometry, free board, and potential lake impacts.

The glacial lakes hazard criteria is based on outburst potential including dam geometry/ free board and risk to the downstream settlements.

Section 3.3

a. The results and methods are not well separated here. The input parameters of the hydrodynamic model fits well in the method section than the results.

The input parameters of the hydrodynamic model including the values assigned is placed in the results section.

b. Section 3.3.1 is not clear how flood volumes were considered for the different GLOF scenarios.

The volume calculations of the lakes are based on area and depth (Refer table 2 of the study). Volume=Area*Depth

c. The potential flood hydro-graphs produced in scenario modeling is not shown. The initial breach hydro-graph is most crucial in any GLOF analysis as it determines the

flood hydraulics downstream as the GLOF propagates along the flow channel. This section needs to be rewritten showing results of the breach parameters and flood hydro-graphs.

The section has been revised.

4. In section 3.3.2 and 3.3.3, the routing parameters are not clear, there has been no mention of the surface roughness along the flow channel.

The routing parameters of the hydrodynamic model including the values assigned are given in table 1 of the study.

5. The boundary condition (upstream and downstream) for routing the potential GLOF event remains unclear.

Refer table 1 of the study.

6. Are the flow depths/ velocities representing the mean value along the cross section or just at a specific point?

The flow depths/ velocities represent the mean value along the cross section at the settlements.

7. There has been no mention of the flood wave arrival timings at specific sites along the flow channel.

Refer table 1 of the study.

Figures:

All figures lack in proper resolution. I recommend exporting the figures with a minimum resolution of 300 dpi for more clarity. Figures 2 and 3 can be combined. The drainage of the basin and the location of the potentially inundated settlements are not shown in the figures. Figure 5 can be removed. Figure 6-The figure lacks location information. The legends remain unclear. The figure can be better represented using other GIS

platforms instead of RAS MAPPER. Figure 5-The figure lacks location information and legends. Figure 9-see comments for figure 6.

Figures 2 and 3 have been combined. The drainage of the basin and the location of the potentially dangerous glacial lakes have been shown in the figure. Overall the figures have been improved as suggested.

References:

Cook, S.J.; Quincey, D.J. Estimating the volume of Alpine glacial lakes. Earth Surf. Dyn. Discuss. 2015, 3, 559–575.

Worni et al., 2013. Glacial lakes in the Indian Himalayas⅘ AĚĞTFrom an area-wide glacial lake inventory to on-site and modeling based risk assessment of critical glacial lakes. Sci. Total Environ. 2013, 468.

Huggel, C.; Haeberli, W.; Kääb, A.; Bieri, D.; Richardson, S. An assessment procedure for glacial hazards in the Swiss Alps. Can. Geotech. J. 2004, 41, 1068–1083.

Sattar, A., Goswami, A., Kulkarni, A., & Emmer, A. (2020). Lake Evolution, Hydrodynamic Outburst Flood Modeling and Sensitivity Analysis in the Central Himalaya: A Case Study. Water, 12(1), 237.

Fujita, K., Sakai, A., Takenaka, S., Nuimura, T., Surazakov, A. B., Sawagaki, T., & Yamanokuchi, T. (2013). Potential flood volume of Himalayan glacial lakes. Natural Hazards & Earth System Sciences, 13(7).

Sattar, A.; Goswami, A.; Kulkarni, A.V. Application of 1D and 2D hydrodynamic modeling to study glacial lake outburst flood (GLOF) and its impact on a hydropower station in Central Himalaya. Nat. Hazards 2019, 97, 535–553.

Froehlich,D.C.PeakouTflowfrombreachedembankmentdam. J.WaterResour. Plan. Manag. 1995, 121, 90–97.

Sattar, A.; Goswami, A.; Kulkarni, A.V. Hydrodynamic moraine-breach modeling and

outburst flood routingâËŸ AËĞTA hazard assessment of the South Lhonak lake, Sikkim. Sci. Total Environ. 2019, 668, 362–378.

Few of them have been inserted.

---

## Author Comment (AC3) · 9 Mar 2020

The following is a list of the specific comments and the changes incorporated in the manuscript:

L18 Why is the water 'turbulent'? Do you mean the ASTER GDEM? If so cite the appropriate version and change in the later text.

Water in glacial lakes is 'turbulent' because of glacial mass movement. The ASTER GDEM has been utilized.

L43 'Indian subcontinent'.

Corrected.

L44 This is not an appropriate reference relating to downstream water supply. See Immerzeel et al. (2010), Quincey et al. (2018), Immerzeel et al. (2020).

Corrected.

L48 Add a citation for 'many disastrous glacial lakes'. 'The damming' of what?.

Inserted. The damming of glacial mass.

L51-52 This is incorrect.

Cited.

L53-55 This needs a citation unless it is related to Che et al, in which case make it clear.

Citation is related to Che et al.

L71-73 This is repeating what has already appeared in the introduction.

Deleted.

L79-80 Also repetition.

Deleted.

L80-81 This sentence on Baltoro is out of place.

Corrected.

L84-85 Surges are not necessarily coupled with glacial lakes.

There is a probability.

L95-97 A lake does not form in all cases.

Not necessarily, but it can happen.

[Figure]

L98 Vital for the stability of what, the lake?

Vital for the stability of the glacier.

L99 What is a 'heavy accumulation zone'?

Heavy accumulation zone, because of heavy snow fall.

L109 quite similar in length'?

The glaciers were quite significant in length in comparison with today.

L111-112 Provide a citation. Is this on average? I'm not sure it is really relevant.

Cited.

L133 Superscript '2'

Corrected.

L152 Provide a citation.

Cited.

L166 I think the grey shading between the panels is unnecessary. What are the brown colored lines within the green polygons? Please show the location of all lakes modeled in this figure.

Figure has been changed.

L176-178 I'm not sure that 'panoramic views' are relevant to the study, nor the following sentence.

Corrected.

L183 Specify the source of the glacier outlines. It would be useful to add an elevation model or river network to show the basin topography and flow direction.

Figure has been changed.

L189 What are the other glaciers in addition to 'valley glaciers'?

Other glaciers in addition to valley glaciers are mountain glacier.

L197 Use degree symbols.

Corrected.

L199 This figure could be added as a panel alongside Figure 2.

Ok.

L205 Is this a version of the ASTER GDEM or a DEM you have generated using ASTER data? How were the data interpolated and why?

ASTER GDEM data has been interpolated to 15m for the purpose of achieving high resolution.

L214-215 Provide an NDWI citation.

NDWI is Normalized Difference Water Index.

L219 Specify the HEC-RAS version you used and whether it was a 1D or 2D model. Refer to other studies that have used HEC-RAS to model GLOFs, including limitations such as consideration of sediment loading and bank erosion.

HEC-RAS version 5.0 is used and it was a 2D model.

L233-234 The meaning of this is unclear.

Deleted.

L256 What is 6b?

It's one of the Landsat-7 band.

L270 What are the specific thresholds used for these decision rules?

If the surface texture is smooth and the slope is not pronounced then such areas are

recognized as glacial lakes.

L279 What is 'erosion and blocked lakes'?

Erosion and blocked lakes are the types of lakes.

L280 Use km2 for consistency.

Corrected.

L285 '5 km'.

Corrected.

L297 Earlier it was stated that you used a 15 m DEM. How was the energy slope derived?

Corrected. Energy slope was derived based on distributing glacial mass flow along boundary condition.

L305 Detail why this empirical relationship is suitable for application to the lakes in your study.

The Costa empirical relationship is suitable for application to the lakes in this study because this empirical formula deals with peak discharge.

L316 Quote fewer significant digits. Does depth refer to a max/mean depth?

Significant digits have been used for depth.

L319 What is the 'data input interval'? Is this the complete duration of the simulation, or time to peak discharge?

Data input interval is the discharge time interval in mins.

L334 Why is this a potentially dangerous lake? The text is not visible on panels d-f. Why does it look like the model formed a large lake on (d)?

The lake is potentially dangerous because of its potential outburst and being hazardous to the downstream settlements.

L343 Why is the damage extent in meters. Shouldn't this be an area?

It is the damage extent in meters.

L347 What was the source of information for the village areas?

The percentage of village habitat was in terms of parcel counts of the village.

L355 The village/buildings should be marked for clarity. Add a scale and north arrow.

Incorporated.

References

Importance and vulnerability of the world's water towers. Nature 577, 364-369. Immerzeel, W.W., Van Beek, L.P.H., and Bierkens, M.F.P. (2010).

Inserted.

---

## Editor Comment (EC1) · Tobias Bolch (Editor) · 25 Mar 2020

Dear Naseem Gilany, dear co-authors,

I have now carefully read the reviews and your replies and also investigated information about academic misconduct. In line with the Copernicus policy and in agreement with the editors in chiefs I decided to reject the manuscript. The reasons are the following:

(1) Both reviewers clearly highlighted several major shortcomings and rated the manuscript quite low. Both also suggested that the manuscript would fit better in a more hazard related journal. The reviews are very much in line with my access review.

I have a lot of respect for the work of reviewers. Both reviewers would have saved time if my recommendations of the access review would have been followed more in depth. I clearly wrote in my access review: "I encourage you to address the points raised above and resubmit the manuscript either to TC or another journal. At minimum I expect to include more relevant literature and improve the structure." My recommendations were neither adequately considered nor the structure really improved. Unfortunately, I provided the opportunity to resubmit without an additional review by me.

(2) The following recently published paper has several similar figures and a similarity report shows almost 50% overlap: Gilany, N., Iqbal, J. Geospatial analysis and simulation of glacial lake outburst flood hazard in Shyok Basin of Pakistan. Environ Earth Sci 79, 139 (2020). This clearly against good scientific practice and also against the Copernicus rules (https://publications.copernicus.org/for_authors/general_terms.html): "The work submitted for publication has not been published before, except in the form of abstracts, preprints, published lectures, theses, discussion papers, or similar formats that have not undergone full journal peer review, and it is not under consideration for peer-reviewed publication elsewhere."

In case you disagree with this decision please contact both the editors in chief and me.

Best regards,

Tobias Bolch - Editor TC

———————————

---

## Author Comment (AC4) · 28 Mar 2020

Dear Editor, Unfortunately, you have not provided the opportunity to resubmit the revised version of manuscript incorporating the changes suggested by both the reviewer. Moreover, the recently published paper, "Geospatial analysis and simulation of glacial lake outburst flood hazard in Shyok Basin of Pakistan" is confined to Shyok Basin only whereas the paper under discussion is the comparative work of both Shyok and Hunza basins. If deemed feasible the revised version of manuscript can be restricted to Hunza basin only. Regards.